# Identifying Faked Responses in Questionnaires with Self-Attention-Based Autoencoders

Alberto Purpura [1,*], Giuseppe Sartori [2], Dora Giorgianni [2], Graziella Orrú [3] and Gian Antonio Susto [1]

1 Department of Information Engineering, University of Padova, Via Gradenigo 6/B, 35122 Padova, Italy; gianantonio.susto@unipd.it
2 Department of General Psychology, University of Padova, Via Venezia 8, 35122 Padova, Italy; giuseppe.sartori@unipd.it (G.S.); dora.giorgianni1@gmail.com (D.G.)
3 Department of Surgical, Medical, Molecular and Critical Area Pathology, University of Pisa, Via Paradisa 2, 56126 Pisa, Italy; graziella.orru@unipi.it
* Correspondence: purpuraa@dei.unipd.it

**Abstract:** Deception, also known as faking, is a critical issue when collecting data using questionnaires. As shown by previous studies, people have the tendency to fake their answers whenever they gain an advantage from doing so, e.g., when taking a test for a job application. Current methods identify the general attitude of faking but fail to identify faking patterns and the exact responses affected. Moreover, these strategies often require extensive data collection of honest responses and faking patterns related to the specific questionnaire use case, e.g., the position that people are applying to. In this work, we propose a self-attention-based autoencoder (SABA) model that can spot faked responses in a questionnaire solely relying on a set of honest answers that are not necessarily related to its final use case. We collect data relative to a popular personality test (the 10-item Big Five test) in three different use cases, i.e., to obtain: (i) child custody in court, (ii) a position as a salesperson, and (iii) a role in a humanitarian organization. The proposed model outperforms by a sizeable margin in terms of F1 score three competitive baselines, i.e., an autoencoder based only on feedforward layers, a distribution model, and a *k*-nearest-neighbor-based model.

**Keywords:** attention mechanism; deep learning; faking detection

## 1. Introduction

In this paper, we address the following unsolved issue: how can we identify when a subject is faking a specific question in a questionnaire? Questionnaires have a widespread usage in the Web and are ubiquitously used to collect information that is usually assumed to be genuine. However, for a variety of reasons, people frequently hide their true responses and the collected data cannot be reliably evaluated. This problem arises because responses to direct questions—such as "Have you ever thought about suicide?"—are easily faked in order to achieve an advantage—e.g., during a job application, applicants may want to hide their emotional instability.

Deception to direct questions may take two different forms: faking-bad and faking-good. Faking-bad characterizes some forensic settings (e.g., criminal, insurance claims) in which the examinee is likely to exaggerate or make up his/her psychological disorder [1,2]. Instead, when faking-good, respondents reduce undesirable aspects to present themselves in more acceptable ways.

Here, we investigate possible solutions for the detection of fake-good responses. These are usually observed in settings in which the respondent is expected to give highly desirable responses. For example, when applying for a job, applicants are highly likely to distort their responses in order to embellish their personality profile (e.g., high integrity, talkative, etc.).

To tackle this faking detection problem, we propose to use an unsupervised deep learning approach based on an autoencoder. An autoencoder [3] is an unsupervised deep learning model that learns to reconstruct its input from a latent hidden representation. Given a set of answers to a questionnaire, we train the proposed model to reconstruct masked responses [4] in the input based on the remaining unmasked ones. To validate the proposed solution, we collect data through a personality test in three different faking conditions (different faking conditions are required as responses are modulated by different objectives).

We ask about 700 volunteers first to take a short personality test faking their responses to achieve a result and specifically (i) to obtain child custody in a divorce case, (ii) to be hired as a salesperson, and (iii) for a generic position in a humanitarian organization. We later asked the same participants to answer the same questions honestly. The questionnaire is a short version of the Big Five personality test proposed in [5], where subjects are asked to provide a numerical estimate with a value between 1 and 5 of their agreement with different statements.

Next, we employ the set of honest answers to the selected personality test to train the proposed autoencoder model. At inference time, we employ the trained model to spot faking patterns in questionnaires by randomly masking each submitted response in the faked questionnaires and comparing the reconstructed value with the faked one. Intuitively, the proposed solution exploits the dependencies and relations between different responses in a questionnaire to spot anomalies in them. Differently from other machine learning models for faking detection, our approach only relies on the sets of honest responses submitted by our volunteers. Indeed, the collection of an exhaustive set of faking patterns would require a much larger-scale data collection effort specific to each faking situation. Instead, we only employ faked responses to evaluate the performance of our model, making the application of the proposed approach in a real-world scenario much more practical than other machine learning models. Our approach is evaluated following the typical evaluation framework of multilabel classification models—where we consider each response in a test as a possible binary label that our model can assign in case of faking—considering the F1 score, precision, and recall metrics that are normally used in this context. We compare the performance of the proposed autoencoder to three different unsupervised baselines, showing how the proposed approach provides a competitive solution for the considered task.

## 2. Related Work

Deception—defined as the action of altering the truth to achieve a certain goal—is a highly relevant topic in different research areas, from Computer Science to Psychology. Deception can happen either between humans [6] or between humans and machines [7,8]. Most research on deception in human interactions focuses on written forms of communications such as dialogues [6], product reviews [9], or dating profiles [10]. In this work, we focus on deception—also known as faking—in the domain of psychological tests.

In psychological questionnaires, faking is usually controlled via the so-called control scales, e.g., the L, F, and K scales of the Minnesota Multiphasic Personality Inventory-2 (MMPI-2) and the X, Y, and Z scales of the Millon Clinical Multiaxial Inventory-III (MCMI-III). These scales detect the respondent's propensity to depict a socially desirable profile or report hyperbolic disorders [11,12]. Abnormal scores in these control scales are taken as evidence of an overall tendency of subjects to alter their responses in the direction of social desirability (fake good) or in the direction of nonexistent psychopathology (fake bad). In short, introverted, hostile and emotionally unstable subjects who describe themselves as extroverted, agreeable, and emotionally stable may be spotted as fakers using such control scales [13].

A number of machine learning approaches were proposed over the years to solve the problem of spotting questionnaires where a subject altered his/her responses [14,15]. In these cases, simple supervised models such as random forests (RF) or support vector

machines (SVM) proved to be able to solve this task with good accuracy. However, to the best of our knowledge, no deep learning procedure was proposed to flag the dishonest responses after a faker profile is identified with the exception of [16]. Indeed, these control scales or machine learning models are only able to detect a general attitude toward the embellishment of the responses, but no method is currently available to flag the specific questions (within the represented questionnaire) that were intentionally altered.

For instance, [17] tested the differences between subjects who reported their true identity and those who faked their identity responding to control, simple, and complex questions. The experiment consisted in an identity verification task, during which response time and errors were collected. Different machine learning models were trained, reaching an accuracy level around 90–95% in distinguishing liars from truth tellers based on error rate and response time. To evaluate the generalization and replicability of these models, a new sample was tested and obtained an accuracy between 80% and 90%. The authors of [18] proposed a method to automatically detect the simulation of depression, which is based on the analysis of mouse movements while the patient is engaged in a double-choice computerized task, responding to simple and complex questions about depressive symptoms. Machine learning models were trained on mouse dynamics features and on the number of symptoms reported by participants. These models reached a classification accuracy up to 96% in distinguishing liars from depressed patients and truth tellers. Despite this, the data were not conclusive, as the accuracy of the algorithm was not compared with the accuracy of the clinicians. Most recently, Ref. [19] confirmed that artificial intelligence performs better than humans in lie-detection tasks based on facial microexpressions.

Furthermore, supervised machine learning models require an extensive data collection phase, where hundreds of volunteer subjects have to provide valid examples of honest responses and faking patterns for the models to learn from. The solution to this problem is not trivial, as faking is a continuous variable and the degree and extent of faking are modulated by the stake and by the strategy under implicit or explicit control of the respondent.

For this reason, in this work, we propose and validate a self-attention-based autoencoder (SABA) to flag faked responses in a questionnaire. An autoencoder is an unsupervised—sometimes also referred as a self-supervised—model which learns to reconstruct its input from a lower-dimensional latent representation. Here, we employ an enhanced version of this model relying on self-attention (SA) [20] to flag faked responses in a questionnaire.

The main advantage of the proposed model is its sole reliance on honest sets of responses, bypassing the need for an exhaustive collection of data related to different faking patterns and, consequently, reducing the cost of employing this model in a real-world scenario. Thanks to this property, responses from public data sources could be employed to train our faking detection model on a popular questionnaire type. If a less popular questionnaire type is employed—for which no public data are available—we believe it would be relatively easy to collect a general set of honest responses by volunteers with no interest in altering their answers to train the proposed approach, compared to a different solution involving the collection of faking patterns or the involvement of experts to develop ad hoc strategies for faking detection.

In addition to this, the proposed model is also capable of producing exact reports of the faked responses in a questionnaire, allowing data collection experts to improve their questionnaires or to adjust their interpretation of the results. Furthermore, relying on local versus global [21] attention patterns, SABA can focus only on the subset of responses in a questionnaire that are useful for the identification of specific faking patterns—compared to traditional machine learning models, which indistinctively rely on all input items.

## 3. Proposed Approach

As we discussed in the above sections, the proposed approach was designed to exploit response correlations patterns to identify faking behavior. More specifically, when we analyze the correlation patterns between different questionnaire items—a more detailed

description of these patterns is provided in Section 5—we observe a stronger correlation between responses when people where asked to alter their answers compared to when they were responding honestly. This can be explained by the fact that people were consistently increasing or decreasing their responses to achieve a certain goal. For example, they were consistently describing themselves as outgoing and extroverted when asked to alter their responses to obtain a job as a salesperson—who is expected to have an extroverted and agreeable personality. Following the above observations, we propose an approach that relies on response correlation patterns to spot faking behavior. This is supported by our intuition that anomalous correlations between responses to certain answers can indicate a systematic behavior aimed at altering the results of a personality assessment conducted through a questionnaire.

**Model Architecture.** The SABA model that we propose for this task is depicted in Figure 1.

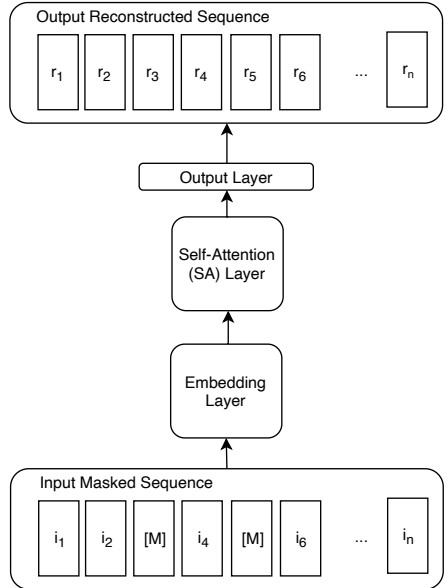

**Figure 1.** Proposed self-attention-based autoencoder (SABA) architecture.

The first layer of our model is an embedding layer, where we assign an embedding vector of size $f$ to each item in a sequence. Specifically, when the model receives a new input sequence $i_1, \ldots, i_n$ (a participant's sequence of responses to the $n = 10$ test items), the corresponding values in the embedding matrix—that are trained as parameters of our model—are rescaled depending on the value of each item in the sequence. Intuitively, this process allows us to consider the responses as words in a sentence and to apply the same SA layer—depicted in Figure 2—as proposed in the transformer model [20] to rescale the input data leveraging on local attention patterns. The SA layer we employ performs the following well-known self-attention formulation [20] on the items of the the input sequence: $SA(Q, K, V) = softmax\left(\frac{QK^T}{\sqrt{f}}\right)V$, where $Q$, $K$, and $V$ are three projections of the representations of the items in the input sequence, obtained through multiplying the input by the respective projection matrices that our model learns. These are typically referred to as query, key, and value representations [20]. Through this operation, we are able to detect high correlations between responses to different questionnaire items. For example, if someone is providing high values as responses to different questions, the product of these values will be associated to high self-attention scores in the respective responses indices pairs. Finally, we employ a feedforward output layer with a sigmoid activation function to estimate the masked input values.

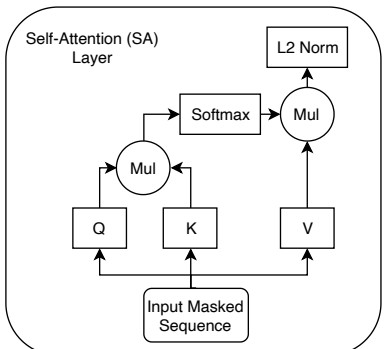

**Figure 2.** Self-attention layer empolyed in the SABA architecture.

**Training Strategy.** The training strategy we employ is similar to the masked language model (MLM) paradigm described in the bidirectional encoder representations from transformers (BERT) paper [4]. We feed a sequence of honest responses to our model, where we mask one or more of them randomly with a special mask token ([$M$]). The model is then trained to predict the values of the responses $r_1, \ldots, r_n$ that were masked in the input sequence, minimizing the mean absolute error (MAE) between the predicted and the true values. Before feeding the data to the model, we normalize all responses from the $[1-5]$ to the $[0, 1]$ range. This normalization allows for more numerical stability during model training and helps keeping the gradients in a manageable range. We train this model only on a collection of honest responses that we collected. The goal of this training strategy is to teach the model to reconstruct the values of each item's response based on the answers the participant gives to nonmasked ones.

**Faking Detection.** We then employ the trained model to flag dishonest responses by iteratively masking each of them and comparing the values our model predicts to the ones each test-taker provided. If the difference between them is larger than a certain threshold, $t$, then we flag this question as faked. Our rationale here is that most responses will follow certain patterns as can be assumed given the intercorrelations between responses reported in Figures 3–5. The model that we train on a set of honest responses learns these correct associations based on the individuals' response styles and allows us to later exploit them for detecting dishonest ones.

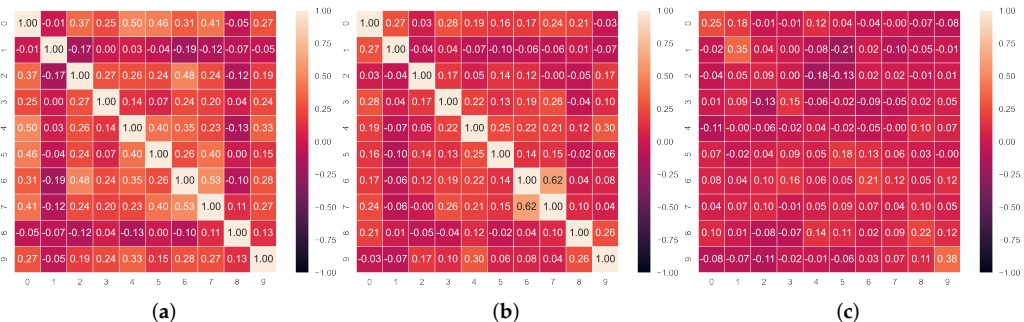

**Figure 3.** Correlation matrices between the honest and/or faked items in the CC dataset. (**a**) Correlation matrix of faked responses. (**b**) Correlation matrix of honest responses. (**c**) Correlation matrix of honest and faked responses.

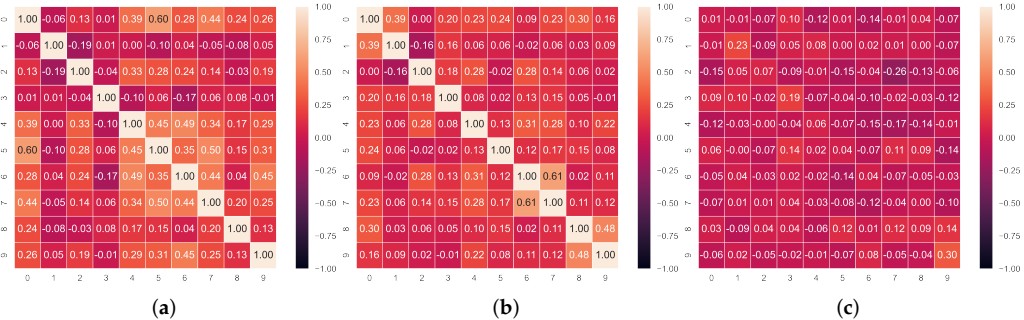

**Figure 4.** Correlation matrices between the honest and/or faked items in the `JIS` dataset. (**a**) Correlation matrix of faked responses. (**b**) Correlation matrix of honest responses. (**c**) Correlation matrix of honest and faked responses.

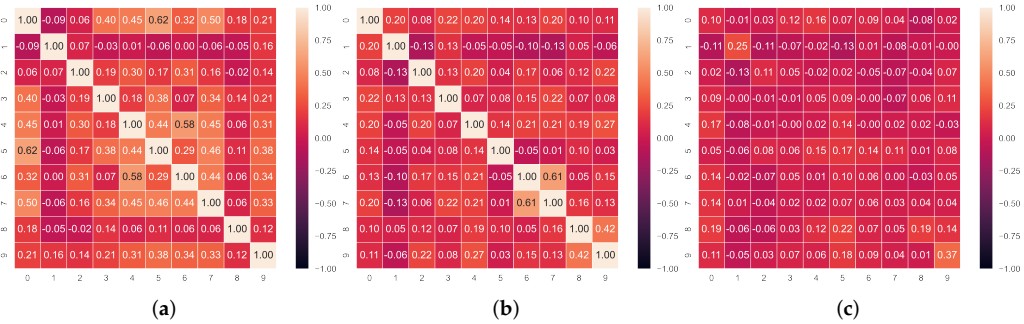

**Figure 5.** Correlation matrices between the honest and/or faked items in the `JIHO` dataset. (**a**) Correlation matrix of faked responses. (**b**) Correlation matrix of honest responses. (**c**) Correlation matrix of honest and faked responses.

The main advantage of this training strategy is that it allows us to train a fake-good detector solely relying on a dataset of honest responses, without the need to collect a comparably large or exhaustive set of faked responses. The faked responses will be used here only to evaluate the performance of the proposed approach.

Furthermore, differently from current psychometric methods, which estimate the abnormality of a response item-by-item—without the information that may be derived from the contextual responses of the same subject to other items of a questionnaire—our model can capitalize on the rich contextual information available to spot faked responses. In other words, we utilize the information from other responses from the same test subject and the correlation information between them to decide whether one or more responses were faked by a person. In addition to this, the model learns the general response and correlation trends from the training dataset of honest responses and tries to apply the learned response paradigms to reconstruct masked responses.

## 4. Materials & Methods

We collected data from a total of 694 participants who responded online to the personality questionnaire described below. The number of participants that took part in each of the three data collection experiments are reported in Table 1. The test subjects in each dataset are distinct and responded to our questionnaire twice, once honestly and once faking their responses. The participants gave their written consent, which was digitally recorded along with their responses to the questionnaire. The participants received no remuneration for taking part in the study. The participants' backgrounds were heterogeneous; for example, in the three studies, the percentage of females was between 75% and 80%, the average age was between 43.1 and 40.8 (with standard deviation between 11 and 14), and the average number of years of schooling was around 15.5 (with standard deviation of around 3).

**Table 1.** Number of participants that took part in each of our data collection experiments. Each of the participants completed the questionnaire twice, the first time responding honestly and the second time altering his/her responses to achieve different objectives.

| Dataset | Participants |
|---|---|
| Child Custody Litigation (CC) | 221 |
| Job Interview, Salesperson (JIS) | 243 |
| Job Interview, Humanitarian Organization (JIHO) | 230 |

This questionnaire is a short version (developed for Web applications) aimed at profiling the personality of the respondent according to the Big Five model [5], mapping the subject responses into five orthogonal dimensions (extroversion, agreeableness, conscientiousness, emotional stability, and openness).

We asked the participants to take our test twice. First, we asked them to answer honestly to all questions and later to take the test pretending to be in one of the following situations when a person would be likely to lie on his/her personality traits: (i) during a psychological assessment to earn the custody of his/her children in a divorce (CC), (ii) during a job interview, to be hired as a salesperson (JIS), and (iii) during a job interview, to be hired by a humanitarian organization (JIHO). Therefore, for each participant, two sets of responses were available: their 10 honest responses and the 10 ones where we asked them to alter their responses. Note that in the latter case, a random number of responses were altered according to the person's personal faking strategy.

For each experiment, we asked three different groups of participants to estimate their agreement using a five-point Likert scale (where 1 indicates a strong disagreement and 5 indicates a strong agreement) to the Italian version of the 10 items from the Big Five Inventory [5,22]. The English version of the statements we considered is reported below [22].
"I see myself as someone who ...":

1. is outgoing, sociable (extroversion);
2. is reserved (extroversion, reversed);
3. tends to find fault in others (agreeableness, reversed);
4. is generally trusting (agreeableness);
5. tends to be lazy (conscientiousness, reversed);
6. does a thorough job (conscientiousness);
7. gets nervous easily (emotional stability, reversed);
8. is relaxed, handles stress well (emotional stability);
9. has an active imagination (openness);
10. has few artistic interests (openness, reversed).

Before our analysis, the responses given to reversed items were complemented so that final scores were all in the same direction.

*4.1. Experimental Setup*

In each of the collected datasets—involving different participants each time—we distinguish between honest and faked responses and train the proposed model only on the honest ones. The goal of our model is to flag the faked responses without any prior knowledge of the participants' response style—as is the case in practical applications.

We evaluate the model's performance comparing each honest and dishonest sequence of 10 responses treating the problem as a multilabel classification approach where each response (1 to 5) is referring to a separate class/question. The metrics we consider are precision, recall, and F1 score. These metrics are formally defined as follows:

$$\text{Precision} = \frac{\text{True Positives}}{\text{True Positives} + \text{False Positives}} \qquad (1)$$

$$\text{Recall} = \frac{\text{True Positives}}{\text{True Positives} + \text{False Negatives}} \tag{2}$$

$$\text{F1 Score} = \frac{2 \times \text{Precision} \times \text{Recall}}{\text{Precision} + \text{Recall}} \tag{3}$$

For example, if the model reported a total of 3/10 responses as faked and two were flagged correctly out of a total of four actually faked responses, then the precision of the model would be equal to 2/3, the recall to 1/2 and the F1 score would be equal to their harmonic mean (4/7).

To avoid overfitting, we train the proposed model on the honest data from two of the three datasets with 10-fold cross validation and early stopping with patience 20—considering the MAE on the reconstruction in the validation set of each fold—and report the average performance the model achieved on the remaining dataset. For example, to compute the performance of the proposed approach on the `CC` dataset, we train our model performing 10-fold cross-validation—considering 10 different training–validation splits—only on the honest responses in the `JIS` and `JIHO` datasets. For each fold, we evaluate the performance of the model on the same test dataset. We report the averaged performance metrics over all folds. The hyperparameters of our model are optimized separately on a random validation set—sampled each time from the training data—considering the following grid of points: number of items to mask during training $n \in \{1, 3, 5\}$, batch size $b \in \{16, 64, 128\}$, feature embedding layer size $f \in \{4, 8, 16\}$, SA layer size $s \in \{4, 8\}$, learning rate $r \in \{0.01, 0.001\}$, and faking detection threshold $t \in \{0.10, 0.15, 0.20, 0.25\}$.

*4.2. Baseline Models*

We also provide different baselines to compare the performance of the proposed approach. All the considered baselines rely on the same training data of the proposed model, i.e., the set of honest responses to the questionnaire our volunteers provided.

**Feedforward MLM Autoencoder.** The first baseline model that we consider is a simplified version of our autoencoder, relying on a simpler feedforward layer instead of an SA one and trained with the same MLM paradigm. The architecture of this model relies on the same embedding layer of the proposed approach, followed by a standard feedforward layer instead of an SA one. The output layer of this model is another feedforward layer. The dimensions of these layers are optimized following the same strategy as the proposed approach described above. We consider this model as a baseline to assess the impact on the final performance of the SA layer.

**Bi-LSTM Model.** The second approach we propose as a baseline is a bidirectional long short-term memory network (LSTM) [3] based model. LSTMs have been a strong modeling solution for sequential data; for this reason, we compare the proposed approach to an unsupervised bidirectional LSTM-based model, trained with the same MLM strategy, relying on a bidirectional LSTM instead of a self-attention model. LSTM models are less efficient types of neural layers compared to SA-based ones that learn contextual representations of items in a sequence by inspecting them in order along a given direction—i.e., left-to-right or right-to-left. A bidirectional LSTM allows us to compute predictions based on the left and right context of the target looking at the input sequence in both directions. The model we employ receives the normalized sequence of responses from our test subjects where some of the items have been masked, passes them through an embedding layer, then through a bi-LSTM and finally through a feedforward layer. The model is trained to predict the masked values in the same fashion as the previously described approaches.

**Distribution Model.** The third approach that we propose as a baseline is a "distribution model" (DM). We propose to compare independently the values associated with each response in a questionnaire to the respective distribution of values in the honest "training" data. To decide whether to flag a response as faked, we compare its value to the one at the $p$th percentile of the responses given to the same question in the honest reference data. The best percentile $p$ we consider in each dataset for faking detection is selected from the

values in the $[0.5, 1]$ range, with a step of 0.05. We start from the 0.5 percentile to avoid flagging a reasonable number of honest responses, i.e., 50% of all the honest responses to each question. We select the percentile value that maximizes the precision metric in the "training" set and then evaluate the approach on a separate test set. The "training" set we employ here is the same we use to train the proposed autoencoder model and the previous baseline.

**$k$-Nearest Neighbor Model.** We also propose a more advanced $k$-nearest-neighbor-based model (NNM). In this case, we hypothesize that faked responses could be spotted by comparing them to the most similar honest questionnaires in a reference set, i.e., the training data we employ for the proposed SABA model. Differently from the DM approach, in this case, we rely on the whole set of responses provided in a questionnaire, with no independence assumption between them. The intuition behind this approach is to find all the "prototypes" of honest responses to a questionnaire that resemble the given sequence of answers to evaluate. For example, if the set of responses to a questionnaire that we are evaluating is similar to one given by one or more of our honest volunteers, we assume the responses he/she provided are honest. Conversely, if 8 out of 10 of the responses are similar to the ones of our honest volunteers while the remaining ones differ by a large margin, then these are likely to have been faked. Following this intuition, for each evaluated questionnaire, we first retrieve the $k$ most similar ones in our honest reference set. The questionnaire similarity metric we employ to compare them is the cosine similarity, a popular measure to compute vector similarity in machine learning [3]. The $k$ value we employ is selected from the $[1, 50]$ range with a step of 5. We then compute for each response in the $k$ honest prototype candidates the average response value to each question and separately compare them to the respective answers in the questionnaire that we have to evaluate. We flag as faked all the responses where the difference with the honest prototype is larger than a certain value $v$—that we optimize on the training data in the $[0, 5]$ range with a step of 1. The values of $v$ and $k$ are optimized to obtain the highest precision in the training set.

For more technical details on the implementation of our approach and baselines, we share the code to compute the results of all our analyses in our online repository (indicated at the end of the article).

## 5. Experimental Results

In this section, we report a statistical analysis on the collected data and the performance evaluation of the proposed SABA model.

### 5.1. Descriptive Statistics

In Table 2, we report the mean and standard deviation of the values associated with different honest and faked responses. From this first set of experiments, we observe that when asked to fake their responses, participants increase their self-evaluation in the direction of social desirability. Conversely, the standard deviation remains similar or lower when considering faked tests compared to honest ones. This is a consequence of the bounded range of values ($[1, 5]$) that we allow as a response. Indeed, for the type of faking we are considering, subjects often have the tendency to answer our questionnaire with values closer to the maximum, inducing a reduction in the observed standard deviation.

Furthermore, we observe that faking patterns differ across datasets. Indeed, we observe a statistically significant difference—$p$-value $< 0.05$ in a paired Student's $t$-test—between the distributions of the honest and faked responses on all items but two—Q6 and Q9—on the CC dataset, and on all items but one in the JIS and JIHO datasets—Q4 and Q6, respectively.

This highlights the presence of different faking patterns for different faking objectives, e.g., efficient faking for child custody is slightly different from efficient faking during a job interview for a salesperson position.

**Table 2.** Mean and standard deviation (SD) of the questions in each honest (H) and faked (F) questionnaire in different datasets. Values marked with a * indicate a statistically significant difference of the faked responses—Student's *t*-test with *p*-value < 0.05—with the honest responses distribution to the same question. Different faking patterns emerge from different situations. For example, Q2 was changed in one direction in CC and in the opposite direction in JIS and JIHO. Average scores to the 10 questions do not differ significantly in the three groups of honest (H) respondents.

| Dataset | Measure | Q1 | Q2 | Q3 | Q4 | Q5 | Q6 | Q7 | Q8 | Q9 | Q10 |
|---|---|---|---|---|---|---|---|---|---|---|---|
| Child custody litigation (CC) | Mean (H) | 3.87 | 2.43 | 2.96 | 3.42 | 3.17 | 4.48 | 2.94 | 3.01 | 3.54 | 3.32 |
| | Mean (F) | 4.47 * | 2.20 * | 3.90 * | 3.69 * | 4.41 * | 4.58 | 4.35 * | 4.23 * | 3.40 | 3.98 * |
| | SD (H) | 0.90 | 0.96 | 0.99 | 0.97 | 1.22 | 0.68 | 1.22 | 1.09 | 1.10 | 1.25 |
| | SD (F) | 0.71 * | 0.93 * | 0.96 * | 0.90 * | 0.85 * | 0.81 | 0.80 * | 0.97 * | 1.24 | 0.93 * |
| Job interview, salesperson (JIS) | Mean (H) | 3.67 | 2.44 | 2.95 | 3.42 | 3.21 | 4.47 | 2.74 | 2.92 | 3.63 | 3.45 |
| | Mean (F) | 4.55 * | 2.75 * | 3.46 * | 3.49 | 4.52 * | 4.67 * | 4.18 * | 4.40 * | 4.12 * | 3.84 * |
| | SD (H) | 0.97 | 1.09 | 1.04 | 1.06 | 1.20 | 0.77 | 1.18 | 1.14 | 1.04 | 1.26 |
| | SD (F) | 0.74 * | 1.25 * | 1.18 * | 1.04 | 0.82 * | 0.74 * | 1.05 * | 0.86 * | 1.06 * | 1.14 * |
| Job interview, Humanitarian Organization (JIHO) | Mean (H) | 3.77 | 2.38 | 3.00 | 3.55 | 3.21 | 4.49 | 2.81 | 3.00 | 3.56 | 3.42 |
| | Mean (F) | 4.53 * | 2.69 * | 3.94 * | 3.96 * | 4.47 * | 4.57 | 4.37 * | 4.35 * | 4.10 * | 4.08 * |
| | SD (H) | 0.95 | 1.04 | 1.11 | 0.96 | 1.25 | 0.71 | 1.16 | 1.18 | 1.10 | 1.26 |
| | SD (F) | 0.86 * | 1.25 * | 1.11 * | 0.94 * | 0.91 * | 0.86 | 0.84 * | 0.88 * | 1.02 * | 0.99 * |

Note that the proposed solution is robust to the variety of faking patterns that might occur in different application scenarios of the same questionnaire. In fact, the aim of our approach is to learn patterns of honest responses. We then detect faking patterns leveraging the reconstruction error of the proposed model when reconstructing a subset of responses in the questionnaire, based on the remaining ones. Thanks to this strategy, we do not need to train our model on an exhaustive set of faking patterns. We require instead on an exhaustive set of honest response types. The number of training samples will depend on the questionnaire length and questions, but according to our experiment, a few hundreds of samples were enough to train an accurate model.

In Figures 3–5, we report the Pearson correlation coefficient between each pair of responses in different faking conditions. From this evaluation, a clear trend emerges. While we observe a high correlation between responses within the honest questionnaires (middle heatmap in each Figure), the correlation between faked responses and corresponding honest ones (rightmost heatmap in each Figure) is virtually nonexistent. The latter result, taken alone, indicates that discriminating the individual responses which underwent distortion is not an easy task, as there is no evident faking pattern among different answers over all our datasets. On the other hand, the correlation between honest responses supports our intuition to rely on an autoencoder model trained with the MLM paradigm. Indeed, the main advantage of an autoencoder is its ability to learn the relations and dependencies between different input features. In our case, we train this model to reconstruct one or more masked response values in a questionnaire. For this reason, observing high correlation patterns between different honest responses to a questionnaire encourages us to believe in the effectiveness of a model which can leverage these patterns (or the lack thereof) to spot faking behaviors.

Furthermore, in Figures 3–5, we observe that overall, the correlation values between faked responses (leftmost heatmaps) are higher than than the ones between honest ones (middle heatmaps). This indicates the presence of a more varied style in honest responses—leading to lower correlation coefficients—compared to the homogeneous style of faked ones—associated to higher correlation coefficients.

*5.2. Performance Evaluation*

In Table 3, we report the F1 score, precision, and recall values of the proposed SABA model and a few baselines. If we compare the performance of the two variants of the proposed neural approach (Table 3), we observe a noticeable improvement across all performance measures when employing an SA layer. Indeed, recalling the previously observed correlations between feature groups, we can expect an SA layer to generate local attention patterns which focus only on the feature groups which are helpful for the

reconstruction of the the masked values with the assumption of an underlying honest response pattern.

**Table 3.** Performance evaluation of the proposed self-attention-based autoencoder (SABA) model trained with the MLM paradigm, the respective version without SA (MLM), a distribution-based model (DM), and a *K*NN-based model (NNM).

|         |      | F1 Score | Precision | Recall |
|---------|------|----------|-----------|--------|
| SABA    | CC   | 0.6335   | 0.6857    | 0.6453 |
|         | JIS  | 0.6650   | 0.7078    | 0.6893 |
|         | JIHO | 0.7280   | 0.6468    | 0.8990 |
| MLM     | CC   | 0.5742   | 0.6288    | 0.5857 |
|         | JIS  | 0.6552   | 0.6751    | 0.6961 |
|         | JIHO | 0.6855   | 0.6484    | 0.8066 |
| Bi-LSTM | CC   | 0.5482   | 0.6707    | 0.5072 |
|         | JIS  | 0.5297   | 0.7091    | 0.4745 |
|         | JIHO | 0.6941   | 0.6496    | 0.8134 |
| DM      | CC   | 0.4057   | 0.4735    | 0.4176 |
|         | JIS  | 0.3413   | 0.4692    | 0.3237 |
|         | JIHO | 0.4142   | 0.4966    | 0.4295 |
| NNM     | CC   | 0.5024   | 0.7262    | 0.4138 |
|         | JIS  | 0.5045   | 0.7548    | 0.4021 |
|         | JIHO | 0.5169   | 0.7279    | 0.4348 |

Compared to the remaining baselines in Table 3, the proposed model achieves a systematically higher precision and recall than the DM model across all datasets, showing how the mutual information between items is a useful predictor for faking. Indeed, we observe that the NNM model—which leverages on the information available from all the responses a subject gives to a questionnaire—achieves a more competitive precision across all datasets, outperforming also the proposed approach. However, its recall values are among the lowest on all datasets, indicating that the model is not very sensitive to small faking patterns in questionnaires and can capture only the most extreme faking behaviors.

Overall, SABA outperforms all the considered baselines when considering the F1 score measure—which promotes a balance between precision and recall—and proves to be the most reliable solution to this problem, avoiding extreme behaviors.

The number of faked responses to each questionnaire can vary widely. To evaluate the robustness of SABA as a function of this number, we report the average F1 score aggregated according to the number of faked responses in a test in Figures 6–8, respectively. We also report the probability density function (PDF) of the number of faked responses per questionnaire in each dataset. From the histograms in Figure 6, we can observe that the performance of SABA is dependent on the number of faked responses to a test. It remains relatively stable if the proportion of faked responses is higher than 60%, but it then decreases almost linearly with the number of faked responses. This is likely due to the model's reliance on the mutual information between different items of a questionnaire. Hence, whenever faking is present, SABA has more difficulties spotting the exact item in the test which was faked and its predictions regarding other responses are also impacted. Despite this, the model achieves a more than satisfying performance when the number of faked responses is between 6 and 8. This is also the most frequent number of faked responses that we observe in all our data if we consider the reported PDF of the number of faked responses per test.

In a real-world scenario, SABA's behavior is desirable, since it indicates a high sensitivity to faking and a low incidence of false negatives. Indeed, the average MAE across different folds on each validation set are 0.20, 0.18, and 0.22, respectively on the CC, JIS and JIHO validation datasets. This indicates that the model is able to reconstruct the response to each question of an honest subject with an average error of less than the resolution of each response value (responses range from 1 to 5, and, when normalized between 0 and 1, they differ by 0.25). There is therefore a high likelihood that honest subjects' responses will not be reported as faked. Conversely, if the answers are not coherent and show patterns that our model never observed, SABA will detect an anomaly and return a set of items that will contain with a high likelihood the faked response/s.

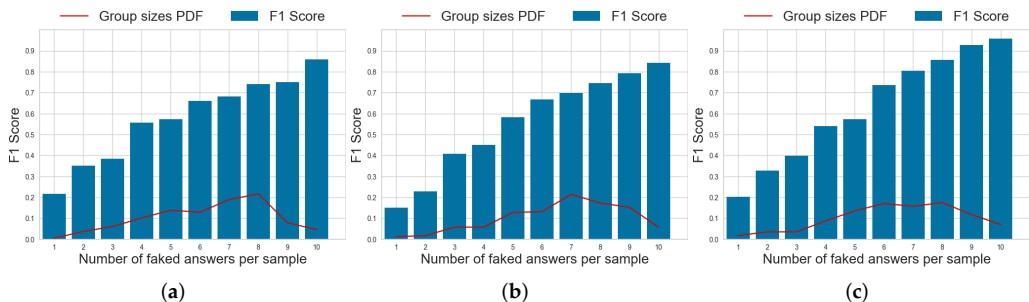

**Figure 6.** Probability density function (PDF) of the number of faked responses per questionnaire and per-item F1 scores for the faking detection task using the SABA model on different datasets. (**a**) dataset CC. (**b**) dataset JIS. (**c**) dataset JIHO.

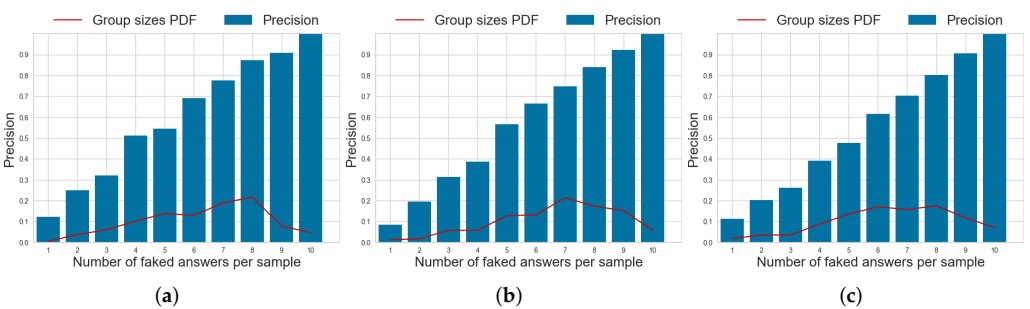

**Figure 7.** Probability density function (PDF) of the number of faked responses per questionnaire and per-item precision for the faking detection task using the SABA model on different datasets. (**a**) dataset CC. (**b**) dataset JIS. (**c**) dataset JIHO.

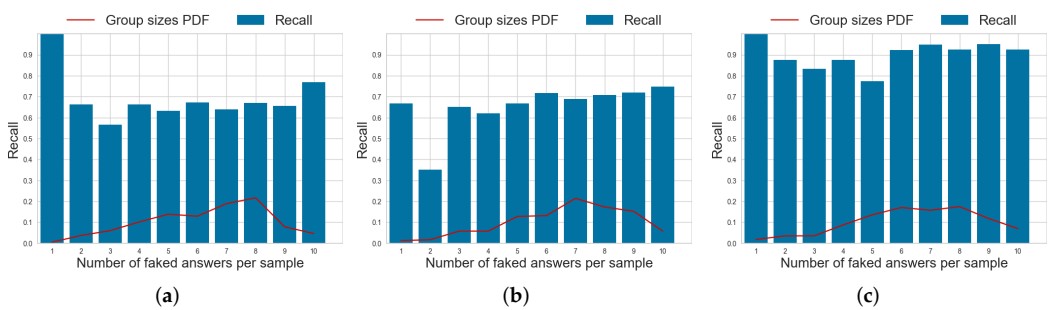

**Figure 8.** Probability density function (PDF) of the number of faked responses per questionnaire and per-item recall for the faking detection task using the SABA model on different datasets. (**a**) dataset CC. (**b**) dataset JIS. (**c**) dataset JIHO.

Indeed, as we observe from the correlation values in Figures 3–5, even if the items in our questionnaire cover orthogonal aspects of a subject's personality (with some redundancy introduced by the reversed control questions), the personality patterns of different people are often correlated, especially the ones emerging from the honest questionnaires. Conversely, if we only consider the correlation coefficients between the responses in faked questionnaires, these are on average lower than the honest ones. This trend can be observed in all correlation matrices associated with our datasets and explains the high impact that just a few faked responses can have on the correlation patterns between responses in a questionnaire on which SABA relies.

## 6. Conclusions and Future Work

In this paper, we addressed the unsolved issue of spotting faked responses in questionnaires. Questionnaires are a de facto solution for collecting information online. However,

in some conditions, when there is an advantage in doing so, people may intentionally alter their responses. Psychometric research developed different techniques—which rely on control scales and/or machine learning models—for spotting when a respondent gave unrealistic desirable responses, but research is virtually absent when it comes to spotting if a specific response has been distorted. To solve this important issue, we developed a novel procedure relying on a self-attention-based autoencoder (SABA) and evaluated its performance in spotting faked questionnaire responses on three different datasets with a total of 694 subjects. The participants were instructed to fake their honest responses to achieve a personal gain in three different settings—a child custody court case and two job interviews.

The main advantages of the proposed model compared to related approaches are (i) its sole reliance on sets of honest responses—versus the need of an exhaustive collection of honest responses and faking patterns—and (ii) the ability to flag single faked responses instead of the entirety of a questionnaire, allowing data collection experts to possibly improve their questionnaires or to adjust their interpretation of the collected results. Furthermore, this research showed that using models that capture a subject's response style to a questionnaire boosts the detection of faked responses compared to models that solely rely for the detection of anomalies on single responses. In our evaluation, SABA outperformed in terms of F1 score all the alternative baseline models we proposed by a sizeable margin. Our experiments also showed how relying on a self-attention (SA) layer and local attention gives a sizeable advantage in recognizing different correlation patterns among the responses as compared to simpler model architectures.

Despite our encouraging results, we believe the described research problem still remains open. For example, pretraining our model on public domain data and fine-tuning it on domain-specific faking behaviors could be one possible option to improve the performance of the proposed solution. Furthermore, we believe that identifying faked responses is a delicate task, and the application of automated approaches to spot this behavior could lead to consequences that could be discussed beyond the scope of this paper. For this reason, we believe more efforts should be devoted also to the development of models that could provide a human-understandable explanation or interpretation for their predictions.

**Author Contributions:** G.S., G.O. and G.A.S. supervised the research work from the substantive and methodological perspective. A.P. and D.G. focused on the data analysis and collection aspects, respectively. All authors have read and agreed to the published version of the manuscript.

**Funding:** This research received no external funding.

**Institutional Review Board Statement:** The study was conducted in accordance with the Declaration of Helsinki, and approved by the Institutional Review Board (or Ethics Committee) of the University of Padua (protocol code n. 4545).

**Informed Consent Statement:** Informed consent was obtained from all subjects involved in the study.

**Data Availability Statement:** Data and code relative to our experiments are available at the following repository: https://github.com/albpurpura/SABA (last visited on 2 March 2022).

**Conflicts of Interest:** The authors have no conflict of interest.

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
