# Peer review of "Identifying Faked Responses in Questionnaires with Self-Attention-Based Autoencoders"

_informatics, doi:10.3390/informatics9010023_

Round 1

Reviewer 1 Report

This article proposed a questionnaire faking detection model, which includes Self-Attention based autoencoders trained with MLM paradigm, and reported an average MAE 0.2(almost all the fake response can be detected).

The article's idea about using unsupervised deep learning approach in the field of deception detection is innovative, but it needs a minor revision.

Below are my specific comments:

  • The correlation between the questionnaire items should be discussed before the approach is proposed, which is the essential basis that MLM can predict masked items effectively.
  • As the materials are collected from real person, information such as consent statement, participants background, remuneration should be provided in the material part.
  • Subplots in Figure 3, 4, 5, 6, 7, 8 should be labeled to make it easier to read.
  • The conclusion includes a summary of the article's work but no discussion about future research and limitations of the research. Write more in this section.

Author Response

Thank you for your positive feedback. The following are our responses to your questions.

? The correlation between the questionnaire items should be discussed before the approach is proposed, which is the essential basis that MLM can predict masked items effectively.
>> We added an additional paragraph at the beginning of Section 3 to explain the relation between the proposed approach and the observed correlations between questionnaire items.

? As the materials are collected from real person, information such as consent statement, participants background, remuneration should be provided in the material part.
>> We added more information about the participants as suggested in the Materials and Methods section.

? Subplots in Figure 3, 4, 5, 6, 7, 8 should be labeled to make it easier to read.
>> The subplots have been labelled and the legends updated.

? The conclusion includes a summary of the article's work but no discussion about future research and limitations of the research. Write more in this section.
>> We developed our conclusions to include more discussions on the limitations of our research and future work opportunities.

Reviewer 2 Report

The paper is an interesting example of autoencoders usage in the identifying faked responses in the psychological questionnaires.

The authors are kindly asked to extend the review of the current works of other authors in the domain, to make the review more comprehensive. There is not enough references regarding to the related works, which are dedicated to the application of Deep Learning and Machine Learning techniques in psychological tests.

The description of the proposed architecture must be improved. What is the content of the input sequence on Figure 1, is it a 10 element vector with participant’s responses in the range 1-5. If it is so, the authors are kindly ask to mention, in the Model Architecture, that n is equal to 10. The Self-Attention mechanism is not sufficiently explained, the Figure 2 should include information, that attention is calculated for each item of the the input sequence.

The formula for calculating the attention SA ( Q, K, V ) is a typical one and it should be underline by the authors, that they apply known formula. Q, K, V should be explained more (query-key-value).

The authors are kindly asked to add general formulas to describe, used in the paper, metrics like: Precision, Recall and F1 Score. Current explanations, based on the example are not enough clear. 

The results are compared only with the methods proposed by the authors. It would be very valuable to present in the paper results and metrics of other authors, which are focusing on similar tasks. The tasks related to implementation of DL in psychology. Lack of such a comparison unables to assess significance of the authors results. 

Author Response

Thank you for your insightful comments, we provide our responses below. Our changes are highlighted in the paper in red.

? The authors are kindly asked to extend the review of the current works of other authors in the domain, to make the review more comprehensive. There is not enough references regarding to the related works, which are dedicated to the application of Deep Learning and Machine Learning techniques in psychological tests.
>> We extended our literature review mentioning more related work to the application of deep learning techniques to the psychological tests literature.

? The description of the proposed architecture must be improved. What is the content of the input sequence on Figure 1, is it a 10 element vector with participant’s responses in the range 1-5. If it is so, the authors are kindly ask to mention, in the Model Architecture, that n is equal to 10. The Self-Attention mechanism is not sufficiently explained, the Figure 2 should include information, that attention is calculated for each item of the the input sequence.
>> We specified in the text the correct value of n that, as indicated is equal to 10, i.e. the number of questions in the questionnaire that we employed in our survey.

? The formula for calculating the attention SA ( Q, K, V ) is a typical one and it should be underline by the authors, that they apply known formula. Q, K, V should be explained more (query-key-value).
>> We added references to the original paper (Vaswani et al.) proposing to employ this formulation for the attention mechanism and provided an explanation of how this attention mechanism helps us finding correlations between different items in input data.

? The authors are kindly asked to add general formulas to describe, used in the paper, metrics like: Precision, Recall and F1 Score. Current explanations, based on the example are not enough clear. 
>> We added definitions for all of the metrics used in section 4.1.

? The results are compared only with the methods proposed by the authors. It would be very valuable to present in the paper results and metrics of other authors, which are focusing on similar tasks. The tasks related to implementation of DL in psychology. Lack of such a comparison unables to assess significance of the authors results. 
>> To the best of our knowledge, we are the first to propose an approach for item-level faking detection. For this reason, we could only compare to other reasonable baselines obtained with comparable modeling strategies to solve the problem we discuss.

Reviewer 3 Report

I like the description of novelty in Section 2, but I did have one question. You haven't explained how a user would acquire training data in the form of true responses. In other words, suppose a company was wanting to administer a questionnaire. While they would not require a training set of faked responses (as you have explained in the text), they would still need a set of true responses specific to that survey, correct? How might this set of questions be acquired? Is there an obvious way of ensuring faked responses are excluded from this true training set?

On line 136, you state the responses were normalized in the [0,1] range. I am a little confused; are you referring to the Big 5 responses that range from 1-5? Was this done for statistical reasons?

Your description of masking selection is a little unclear. On line 139, you describe teaching the model to reconstruct response values based on non-masked values. Does this imply that none of the non-masked responses were faked (since you are only training on true responses)? If so, how did you identify the subset of responses that were candidates for masking? Did this include the set of all faked responses combined with a decoy set of true responses?

Your statement on line 144, that a person who provides a low answer is not likely to provide a high answer to a different question, essentially forms the foundation for your entire methodology. Do you have any supporting citations to validate this claim? It seems logical, but since it is so pivotal to your proposed technique, I suggest supporting it with a citation.

What do you mean by 'rich contextual information' on line 155? Can you elaborate further? Are you describing the pattern of a respondent's behavior or something else?

What is the overlap in Table 1? In other words, how many participants took all three surveys? (I assume it is less than 221) Were you able to use data from respondents who only took one of the questionnaires or does this create issues when validating your approach? Wouldn't you need them to respond to all three surveys in order to provide a valid assessment of your technique?

On line 170, you refer to the sets of responses as the 10 honest responses and the 10 faked responses. However, not every question was faked in the second set, correct? The participants were told to act as if they were in a certain situation (e.g., applying for a job), but they weren't instructed to lie on every question, were they? What were the exact instructions given to the participants?

The paper includes a few run-on sentences that should be shortened. For example, lines 196-201 are written as a single sentence.

I recommend expanding your discussion on line 282. It is evident from the results that different faking patterns emerged for each survey (as expected). How might this impact the robustness of your model for broad application, since no cross-set faking patterns emerged? (line 289) Do you anticipate the faking patterns across thousands of different surveys to be sufficiently consistent that your model could be applied to all of them? If so, what is the basis of this assumption? This would be worth addressing.

The first sentence in Section 5.2 needs to be revised.

Author Response

Thank you for your insightful comments, we provide our responses below. Our changes are highlighted in the paper in red.

? I like the description of novelty in Section 2, but I did have one question. You haven't explained how a user would acquire training data in the form of true responses. In other words, suppose a company was wanting to administer a questionnaire. While they would not require a training set of faked responses (as you have explained in the text), they would still need a set of true responses specific to that survey, correct? How might this set of questions be acquired? Is there an obvious way of ensuring faked responses are excluded from this true training set?
>> Thank you for your positive feedback. We briefly expanded that section to discuss possible options for data collection. For example, responses from public data sources could be employed to train our faking detection model on a popular questionnaire type. If a less popular questionnaire type is employed -- for which no public data are available -- we believe it would be relatively easy to collect a general set of honest responses by volunteers with no interest in altering their answers to train the proposed approach, compared to a different solution involving the collection of faking patterns or the involvement of experts to develop ad-hoc strategies for faking detection.

? On line 136, you state the responses were normalized in the [0,1] range. I am a little confused; are you referring to the Big 5 responses that range from 1-5? Was this done for statistical reasons?
>> What you observed is correct, we normalize the values from 1-5 to the 1-5 range. We apply this normalization to help numerical stability during model training and to keep model gradients and weights within standard intervals. We added a discussion of this strategy in the paper.

? Your description of masking selection is a little unclear. On line 139, you describe teaching the model to reconstruct response values based on non-masked values. Does this imply that none of the non-masked responses were faked (since you are only training on true responses)? If so, how did you identify the subset of responses that were candidates for masking? Did this include the set of all faked responses combined with a decoy set of true responses?
>> Only honest responses are employed at training time. We mask a few of these values and then train our model to reconstruct them. The subset of responses that are masked are selected randomly. Faked responses are not included in the training set because our goal is to learn to reconstruct non-anomalous data. We then employ our model to attempt at reconstructing responses as if they were honest. If we see that the model's reconstruction differs from the actual value of the response, then we flag this as faked. We specified this better in the revised version of the paper.

?  Your statement on line 144, that a person who provides a low answer is not likely to provide a high answer to a different question, essentially forms the foundation for your entire methodology. Do you have any supporting citations to validate this claim? It seems logical, but since it is so pivotal to your proposed technique, I suggest supporting it with a citation.
>> Our rationale here is that most responses will follow certain patterns as can be assumed given the intercorrelations between responses reported in fig. 3 and 4. We reformulated that statement in the paper.

?  What do you mean by 'rich contextual information' on line 155? Can you elaborate further? Are you describing the pattern of a respondent's behavior or something else?
>> We included a paragraph in the paper explaining better what we mean with rich contextual information and how that is related to the inner workings of the proposed approach.

? What is the overlap in Table 1? In other words, how many participants took all three surveys? (I assume it is less than 221) Were you able to use data from respondents who only took one of the questionnaires or does this create issues when validating your approach? Wouldn't you need them to respond to all three surveys in order to provide a valid assessment of your technique?
>> The participants to each experiment are all different. There was no overlap. We added a statement in the paper specifying this. Also, our experiments in the three proposed scenarios are independent for what concerns faked responses. On the other hand, if we consider only honest responses, the questionnaire responses from different subjects would be interchangeable since they refer to the same questionnaire. However, we considered them separately for evaluation purposes since we had to compare honest and faked responses from the same person. We further specified this in the paper.

? On line 170, you refer to the sets of responses as the 10 honest responses and the 10 faked responses. However, not every question was faked in the second set, correct? The participants were told to act as if they were in a certain situation (e.g., applying for a job), but they weren't instructed to lie on every question, were they? What were the exact instructions given to the participants?
>> That is correct, participants were not lying to every question. They were free to pick which questions to lie to. We corrected this statement in the paper specifying what we just mentioned. Thank you for pointing out this imprecision. 

? The paper includes a few run-on sentences that should be shortened. For example, lines 196-201 are written as a single sentence.
>> We revised the paper and improved the presentation of a few sentences like the one at lines 196-201.

? I recommend expanding your discussion on line 282. It is evident from the results that different faking patterns emerged for each survey (as expected). How might this impact the robustness of your model for broad application, since no cross-set faking patterns emerged? (line 289) Do you anticipate the faking patterns across thousands of different surveys to be sufficiently consistent that your model could be applied to all of them? If so, what is the basis of this assumption? This would be worth addressing.
>> Thank you for the very relevant suggestion. We expanded the indicated paragraph with more information on what we observed in our experiments.

? The first sentence in Section 5.2 needs to be revised.
>> We reformulated that sentence.

Reviewer 4 Report

The authors combine machine learning-based models with the practical research question of faked answers to questionnaires, which is an important practical problem. Whist the overall F1 score in identifying faked answers is not yet in the range of generally being able to remove faked answers, they show that the model significantly improves against a baseline. I think the research topic, as well as approach and results, are interesting and practically relevant.

Here are some observations for possible points of improvement I had for your paper:

  • You argue that the problem of classifying false responses to questionnaires is mainly an unsupervised learning problem since one does not usually know whether an answer is true or fake, but has access to a lot of questionnaire responses. This makes sense, but the dataset you collect does not seem to be one that is mainly unsupervised (all participants give true and faked answers). It would be great to show that your method also works if one pre-trains the models on a large dataset of answers that were generally given to these types of surveys and are publicly available. This is the data an organization could also have access to. It would be a good sign if your model gets even better if such data that was not generated in your experiment is additionally mixed in and improves the results.
  • I can see your attention-based model is a good choice, but why not take the traditional workhorse of sequential learning, a Bi-Directional- LSTM as a baseline as well. The LSTM could also be trained to predict the answer for a particular question given the answers to other questions. It would be interesting to see what type of model performs better.
  • Since your data is not dominated by the “unsupervised” data it would be interesting to see how a model such as an LSTM performs that directly predicts whether an answer is true or fake, given a sequence of previous answers. Does directly including this supervision signal improve or deteriorate performance? Ultimately, if it does not improve performance this would really make a stronger point for unsupervised methods.
  • Could you also report AUC in addition to precision, recall and F1

All the best of luck with your paper.

Author Response

Thank you for your insightful comments, we provide our responses below. Our changes are highlighted in the paper in red.

? You argue that the problem of classifying false responses to questionnaires is mainly an unsupervised learning problem since one does not usually know whether an answer is true or fake, but has access to a lot of questionnaire responses. This makes sense, but the dataset you collect does not seem to be one that is mainly unsupervised (all participants give true and faked answers). It would be great to show that your method also works if one pre-trains the models on a large dataset of answers that were generally given to these types of surveys and are publicly available. This is the data an organization could also have access to. It would be a good sign if your model gets even better if such data that was not generated in your experiment is additionally mixed in and improves the results.
>> We agree with this suggestion, indeed, we only employed faked response pairs to evaluate the effectiveness of the proposed approach and not to improve its performance. It would therefore be useful to employ publicly available data from the Web to improve the training of the proposed approach. We evaluated the performance of the proposed strategy when using a dataset from https://openpsychometrics.org/_rawdata/ (Answers to the Big Five Personality Test, constructed with items from the International Personality Item Pool) for data augmentation during the unsupervised training of the proposed approach -- we selected only a subset of the responses available to obtain a set of questionnaire responses of the same size of the data we already collected -- but did not observe any performance improvement. This is likely due to the lack of an exact correspondence between the data we collected and the data available online and the differences in how the data were collected. We added a paragraph to discuss this option as future work in the conclusions section.

? I can see your attention-based model is a good choice, but why not take the traditional workhorse of sequential learning, a Bi-Directional- LSTM as a baseline as well. The LSTM could also be trained to predict the answer for a particular question given the answers to other questions. It would be interesting to see what type of model performs better.
Since your data is not dominated by the “unsupervised” data it would be interesting to see how a model such as an LSTM performs that directly predicts whether an answer is true or fake, given a sequence of previous answers. Does directly including this supervision signal improve or deteriorate performance? Ultimately, if it does not improve performance this would really make a stronger point for unsupervised methods.

>> We agree with your suggestion, an LSTM is a popular modeling strategy that should be included in our baselines. We added a new set of experiments including a Bi-LSTM-based model. The Bi-LSTM is employed here as an alternative to the Self-Attention layer in the proposed approach. We believe comparing the proposed unsupervised approach with a supervised one would be unfair to evaluate our modeling strategy and would yield an advantage to any supervised model able to leverage on larger efforts/investments on data collection. The main advantage of our unsupervised approach is to take advantage of unlabelled data for the training of our model, making it cheaper and easier to employ in a real-world scenario. Moreover, training a model in a supervised fashion would limit the amount of training data we could use. Currently, we employ the honest responses collected from all participants to our experiments combined to train the proposed model and baselines. If we had to train a supervised approach for faking detection, we would be only able to employ about 200 honest/faked responses pairs for the training of each model (on each dataset CC, JIHO and JIS) since the faking patters would differ across different experiments and data would not be shareable/reusable across them. 

For this reason, even if we agree with the reviewer that in principle a supervised model would outperform the proposed unsupervised one if the appropriate amount of training data would be available, we would not have enough data (only a couple hundred sets of responses) to train it properly and to perform a fair comparison. Therefore, we only include a Bi-LSTM based model trained in an unsupervised fashion to the baselines to compare the proposed modeling strategy to an additional popular solution for modeling sequential data.

? Could you also report AUC in addition to precision, recall and F1
>> We believe that AUC would not be the best performance measure to compare the proposed approach with the considered baselines. In fact, the distribution model and the k-NN approach we propose to employ as baselines, are approaches that are not fit for this evaluation. For what concerns the proposed distribution model, setting a different threshold would mean changing the model itself since the approach we proposed is founded on proposing a strategy to select a certain threshold value for faking detection. Similarly, the only parameter that could be comparable to a threshold in the k-NN model is the number of neighbors employed in the classification. Changing this parameter means changing the model itself and would yield non-comparable results to the previous approaches. Moreover, the k-NN model provides only a discrete classification which is not dependent on any threshold for the interpretation of its outcome. For this reason, we employ the proposed evaluation measures instead of AUC for the evaluation of the proposed approach.